# Navigating Cognitive Manifolds: Optimal Transport for Large Language Model Optimization

## Abstract

Large language models (LLMs) possess vast knowledge but face inefficiencies in task-specific knowledge organization and activation. Existing prompt engineering relies on empirical trial-and-error, lacking principled optimization frameworks. We introduce Cognitive Geometry Optimal Transport (CGOT), a framework that reframes LLM cognitive optimization as geometric navigation in high-dimensional probability spaces. Our key insight models cognitive configurations as probability measures over knowledge states, leveraging optimal transport theory to derive principled paths from initial to target configurations. CGOT employs a dual geometric guidance system: Wasserstein distances for radial metrics and Kantorovich potential gradients for directional guidance, enabling continuous optimization on cognitive manifolds. Through systematic experiments on three prominent LLMs (Qwen3-72B, Deepseek-v3-67B, LLaMA-3-70B) across four cognition-intensive benchmarks (GSM8K, HumanEval, CommonsenseQA, BigBench-Hard), we demonstrate: (1) LLM cognitive spaces exhibit low-dimensional manifold structures (intrinsic dimension 8.7) with strong geometry-performance correlation (Pearson $r = -0.76$, robustified to standardized $\beta = -0.82$ under hierarchical mixed-effects modeling); (2) CGOT achieves consistent 4.8% average performance gains (Cohen's d $> 0.7$ in structured tasks), outperforming baselines like APO, OPRO, GrIPS, and BayesOpt-Prompt by 0.6% on average (p<0.05); (3) the framework generalizes across prompt strategies (Zero-shot: +5.3%, Few-shot: +4.5%, Chain-of-Thought: +4.6%) and model architectures. Ablation studies confirm the critical contributions of Wasserstein metrics (-1.3% without) and non-linear optimization (-2.2% without). This work bridges optimal transport theory with LLM optimization, transforming prompt engineering from empirical art to geometric science with enhanced process interpretability.

## 1 Introduction

The fundamental challenge in large language models lies not in the scale of knowledge, but in the *organization* of knowledge. While large language models demonstrate exceptional performance in explaining algorithmic principles, they often fail to reliably execute these algorithms—a phenomenon referred to as the "understanding-execution separation" Zhang (2025); Bommasani (2021). This performance bottleneck stems from the difficulty of efficiently organizing and activating task-relevant knowledge subsets from vast knowledge repositories Mitchell (2021). This challenge can be modeled as a **cognitive space optimization problem**—namely, controlling the activated knowledge states in cognitive space to form the optimal cognitive configuration required for tasks Binz & Schulz (2023). Here, cognitive space refers to the high-dimensional manifold constituted by all possible activation patterns in the model's parameterized hidden state representations; knowledge states are local activation configurations on the manifold, typically manifested as hidden states in Transformers. Research demonstrates that hidden states encode task-specific semantics, syntax, and reasoning pathways Valeriani et al. (2023); Ansuini et al. (2019)(Fig. 1). These findings support the validity of hidden states as proxies for knowledge states Saxe et al. (2019).

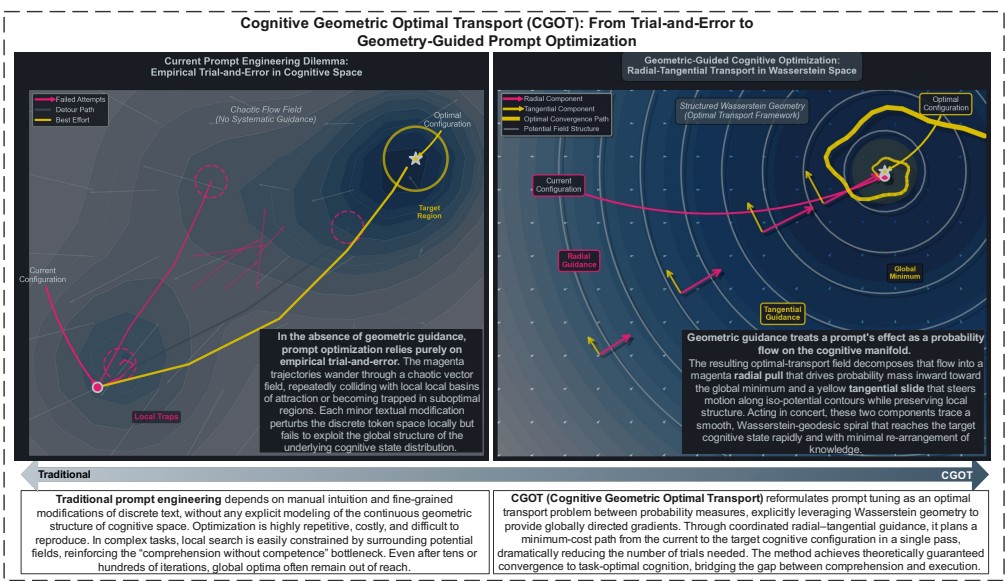

Figure 1: **Cognitive Geometric Optimal Transport (CGOT): From trial-and-error to geometry-guided optimization.** (Left) Without geometric guidance, prompt optimization relies on trial-and-error search, often trapped in suboptimal regions. (Right) CGOT formulates prompt tuning as an optimal transport problem, decomposing probability flow into radial pull and tangential slide. This yields smooth Wasserstein-geodesic paths to target cognitive states with minimal knowledge rearrangement.

We argue that current prompt engineering approaches remain largely an "empirical art." They rely on empirically-driven local constraint strategies, lacking systematic theoretical frameworks for global optimization Wei et al. (2022); Zhou et al. (2022). Mainstream methods such as few-shot learning and chain-of-thought reasoning are essentially "undirected constraint mechanisms," relying on discrete trial-and-error adjustments (or heuristic search), which are inefficient when facing complex tasks Bommasani (2021). To transition from this "empirical art" to a "geometric science," the key insight is that **cognitive states exhibit continuous distributional properties**. Evidence from multiple research domains indicates that cognitive states possess continuous distributional characteristics: neuroscience research reveals that cortical cognitive representations are encoded in continuous neuronal cluster activation patterns Mitchell (2021); conceptual space theory demonstrates the continuous distribution and similarity structures of conceptual knowledge in geometric spaces Zou et al. (2025b); Saxe et al. (2019); representation learning research reveals the continuous geometric structure of semantic knowledge through word embeddings and neural network hidden layer analysis Valeriani et al. (2023); computational cognitive science further confirms the continuous dynamical characteristics of cognitive state transitions Binz & Schulz (2023). From a mathematical perspective, the cognitive process in LLMs can be understood as probability distribution changes in high-dimensional knowledge state spaces Ansuini et al. (2019). Each prompt activates specific subsets of knowledge states, forming the cognitive configuration under that task—namely, the probability measure of knowledge states. The essence of optimizing cognitive configurations is finding the optimal transformation path from the current probability distribution to the target probability distribution.

Optimal transport theory provides an elegant mathematical framework for this challenge. By modeling current and target cognitive configurations as probability measures, cognitive optimization transforms into the problem of finding optimal transport plans between measures Peyré et al. (2019). The Wasserstein distance quantifies the "true distance" between cognitive configurations, while Kantorovich potential functions construct continuous "cognitive potential fields," providing clear gradient directions toward optimal targets Santambrogio (2015).

Based on this foundation, this paper introduces the **Cognitive Geometric Optimal Transport (CGOT) framework**. CGOT precisely characterizes optimization paths through dual geometric information: Wasserstein distance provides radial metrics, while Kantorovich potential function gradients provide directional guidance. This approach fundamentally reconceptualizes prompt optimization: instead of a discrete search for magic words, it becomes a continuous geometric navigation problem on the cognitive manifold. This achieves a fundamental transition from "empirically-driven local constraint adjustment" to "radial-directional guided geometric optimization."

The main contributions of this work are summarized as follows:

- **Paradigm Shift:** We introduce optimal transport theory to formalized LLM cognitive optimization, bridging the gap between discrete empirical prompt engineering and continuous, theoretically guided geometric optimization.
- Develop a probabilistic manifold geometry approach for cognitive space modeling, uncovering the intrinsic geometric principles of knowledge state configurations;
- Propose the CGOT framework to enable precise control and efficient optimization of cognitive configurations via a rigorous geometric navigation system;
- **Systematic Validation:** We provide comprehensive empirical evidence (including a robust negative association of $\beta = -0.82$ between Wasserstein distance and performance validated via mixed-effects modeling) that confirms the geometric hypothesis. Our results across multiple benchmarks demonstrate that this geometric paradigm yields consistent performance gains over heuristic baselines.

## 2 RELATED WORK

### 2.1 OPTIMAL TRANSPORT THEORY AND NEURAL REPRESENTATION GEOMETRY

Optimal transport (OT) theory has transitioned from mathematics to a key tool in machine learning. In generative modeling, Wasserstein GANs (WGANs) use OT distances to resolve vanishing gradients in traditional GANs Arjovsky et al. (2017). Recent applications include RoPE for addressing model miscalibration as an OT problem Wehenkel et al. (2024), Meta Flow Matching for efficient probabilistic modeling via Wasserstein manifold vector fields Atanackovic et al. (2024), Tree-Wasserstein distances for multi-scale hierarchical feature learning Lin et al. (2024), and the identification of "Wasserstein neurons" in LLMs linked to polysemy and sparsification Sawmya et al. (2024).

Concurrently, neural representation geometry research uncovers structural patterns in deep models, such as the "expansion-contraction-stabilization" evolution in Transformer representations Valeriani et al. (2023). Geometric deep learning highlights symmetry and structure preservation as core encoding principles Bronstein et al. (2021), while topological deep learning captures higher-order relations using structures like simplicial complexes Hajij et al. (2022); Zou et al. (2025a); Papamarkou et al. (2024). Cognitive interpretability (CogInterp) systematizes explanations of high-level cognitive processes in DL models Thomas et al. (2022).

Despite progress, integrations of OT and neural geometry for optimizing cognitive spaces in LLMs are rare, often overlooking probability manifold geometry for global optimization.

### 2.2 PROMPT OPTIMIZATION AND COGNITIVE CONTROL

Prompt optimization enhances LLM performance through manual or automated approaches. Manual techniques include Chain-of-Thought (CoT) for improved reasoning Wei et al. (2022) and Self-Consistency for reliability via multi-path sampling Wang et al. (2022). Automated methods aim to minimize human input, such as PromptAgent's strategic planning for expert-level prompts Wang et al. (2023), Robust Prompt Optimization (RPO) for adversarial robustness Zhou et al. (2024), and Concentrate Attention for domain generalization Li et al. (2024). Recent advancements in automated prompt optimization further include EASE for efficient ordering-aware selection of exemplars Wu et al. (2024), efficient optimization via best arm identification Shi et al. (2024), interpretable prompt optimization for vision-language models Du et al. (2024), localized zeroth-order prompt optimiza-

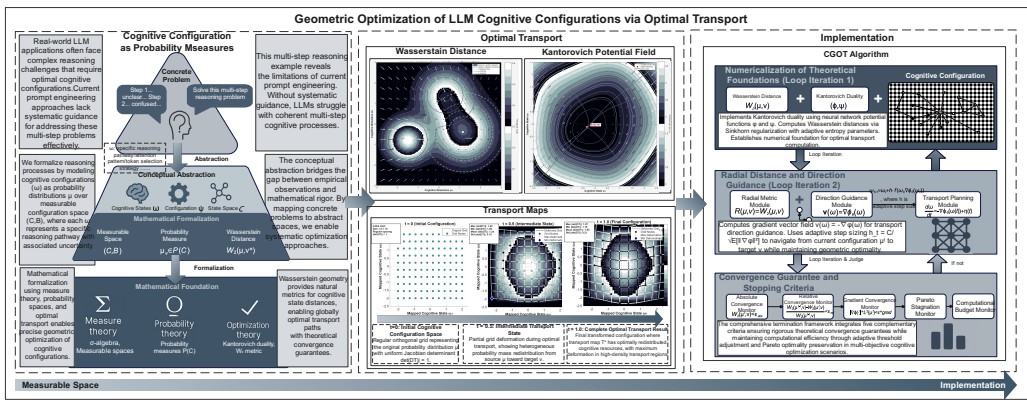

Figure 2: **Geometric optimization of LLM cognitive configurations via optimal transport.**
The framework maps concrete problems to mathematical formalization, modeling cognitive states
as probability measures over measurable spaces. Wasserstein geometry defines state distances, en-
abling the CGOT algorithm to plan optimal transport with radial and tangential guidance. Iterative
updates move probability mass from source to target efficiently, ensuring convergence and reducing
cognitive optimization cost.

tion Hu et al. (2024), and the integration of large language models with evolutionary algorithms to
create powerful prompt optimizers Guo et al. (2023).

Nevertheless, these methods remain empirical, lacking theoretical explanations for prompt efficacy.
They view optimization as discrete operations, ignoring continuous knowledge activation distribu-
tions on probability manifolds and cognitive space geometry, resulting in inefficient local searches
and challenges in achieving global optima.

## 3 METHODS

This section outlines our proposed framework for optimizing the cognitive configuration of large
language models (LLMs), drawing from optimal transport theory. We begin by formalizing the
cognitive mechanisms and then detail the geometric approach used for optimization. As illustrated
in Figure 2, our framework transforms the complex task of prompt engineering into a mathematically
tractable problem of geometric optimization.

### 3.1 FORMAL MODELING OF COGNITIVE CONFIGURATION OPTIMIZATION

This section formalizes the cognitive mechanisms of large language models into a mathematical
framework, establishing optimization mappings from current cognitive states to target cognitive
states.

#### 3.1.1 PROBABILITY MEASURE REPRESENTATION OF COGNITIVE SPACE

**Core Idea**: Model the cognitive mechanisms of LLMs as optimization problems on probability
measure spaces.

We define the cognitive space as a measurable space $(\mathcal{C}, \mathcal{B})$, where $\mathcal{C}$ represents the set of all possible
knowledge states. The cognitive space can be decomposed into a Cartesian product of knowledge
components: $\mathcal{C} = \prod_{i=1}^{n} \mathcal{K}_i$

where $\mathcal{K}_i$ represents the state space of the $i$-th knowledge component.

**Probability Representation of Cognitive Configuration**: Given prompt configuration $\psi$, the
model's cognitive configuration is represented as a probability measure $\mu_\psi \in \mathcal{P}(\mathcal{C})$, characteriz-
ing the activation probability distribution of various knowledge states.

**Computational Implementation**: To make the theoretical framework computationally tractable, we use model hidden state features as proxies for cognitive states. The empirical estimation process is detailed in Algorithm 1.

**Knowledge State Activation Modeling**: We employ a hierarchical structure to model knowledge activation probabilities: $p(\omega|\psi) = \prod_{i=1}^{n} p(\omega_i|\psi, \mathrm{pa}(\omega_i))$, where $\mathrm{pa}(\omega_i)$ denotes the parent nodes of knowledge component $\omega_i$, reflecting knowledge dependency relationships.

### 3.1.2 OPTIMIZATION PROBLEM FORMALIZATION

The cognitive space optimization problem aims to find an optimal transport map $T : \mathcal{C} \to \mathcal{C}$ such that the current cognitive configuration $\mu$ approaches the target configuration $\nu$ (which is empirically estimated from a small set of high-quality "gold" examples, see Appendix A.1) as closely as possible after transport, while satisfying multiple constraints to ensure validity. In this context, our framework functions as a geometric navigation system that plans the optimal path to the target $\nu$, distinguishing it from heuristic search methods that struggle to locate the destination. This optimization process must satisfy multiple constraints to ensure that the generated cognitive configuration maintains both the model's inherent consistency and effective adaptation to target tasks. These constraints include:

- **Cognitive Consistency**: To preserve the structural invariance of core knowledge relationships.
- **Task Relevance**: To ensure the optimization direction aligns with the target task.
- **Computational Feasibility**: To limit the overhead of the optimization process.

Combining these constraints, the cognitive configuration optimization problem is formulated as:

$$\min_{T \in \mathcal{T}} W_2(\mu, \nu) + \lambda_1 \mathcal{L}_{\text{consistency}} + \lambda_2 \mathcal{L}_{\text{task}} \tag{1}$$

where $W_2$ is the 2-Wasserstein distance. The specific mathematical formalizations of these constraints and the complete framework are provided in A.2. This formalization provides the mathematical foundation for our subsequent solution via Kantorovich duality.

### 3.2 COGNITIVE GEOMETRY THEORY BASED ON OPTIMAL TRANSPORT AND CGOT FRAMEWORK

This section integrates cognitive geometry theory with the CGOT framework implementation, avoiding redundancy, and unfolds logically from theoretical foundations to algorithmic realization: first establishing the Wasserstein geometric structure, then constructing Kantorovich potential functions, followed by introducing manifold representation and radial-directional guidance system, and finally presenting the iterative optimization algorithm.

### 3.2.1 WASSERSTEIN GEOMETRIC STRUCTURE OF COGNITIVE CONFIGURATION SPACE

**Core Idea**: Establish geometric measures between cognitive configurations to provide mathematical foundations for optimization.

Based on the probability measure representation from Section 5.1, we introduce the Wasserstein distance as the core metric between cognitive configurations. For two probability measures $\mu, \nu \in \mathcal{P}(\mathcal{C})$ on cognitive space $(\mathcal{C}, d)$, the p-Wasserstein distance is defined as:

$$W_p(\mu, \nu) = \left( \inf_{\gamma \in \Pi(\mu, \nu)} \int_{\mathcal{C} \times \mathcal{C}} d(\omega, \omega')^p \, d\gamma(\omega, \omega') \right)^{1/p} \tag{2}$$

where $\Pi(\mu, \nu)$ is the set of all coupling measures with marginals $\mu$ and $\nu$, and $d(\omega, \omega')$ characterizes the semantic distance between knowledge states.

**Manifold Structure of Cognitive Space**: Cognitive states are embedded in a low-dimensional manifold $\mathcal{M} \subset \mathbb{R}^D$. By constructing local neighborhood graphs and defining similarity weights $w_{ij} = \exp(-\|\mathbf{h}_i - \mathbf{h}_j\|^2 / 2\sigma^2)$, we use spectral embedding or diffusion maps to extract intrinsic

coordinate systems of the manifold. The geodesic distance $d_{\mathcal{M}}(\omega, \omega')$ on the manifold replaces Euclidean distance as the base metric for Wasserstein distance.

**Otto Geometry Application**: Based on the manifold structure, the Wasserstein space $(\mathcal{P}(\mathcal{M}), W_2)$ forms an infinite-dimensional Riemannian manifold whose tangent space is spanned by gradient fields, providing theoretical foundations for geometric analysis and optimization path planning of cognitive configurations.

### 3.2.2 COGNITIVE POTENTIAL FIELD CONSTRUCTION VIA KANTOROVICH POTENTIAL FUNCTIONS

**Core Idea**: Construct potential functions through duality theory to provide "directional guidance" in cognitive space.

For current cognitive configuration $\mu$ and target configuration $\nu$, the Kantorovich dual form is:

$$W_1(\mu, \nu) = \sup_{\phi \in \Phi} \left\{ \int_{\mathcal{C}} \phi(\omega) \, d\mu(\omega) - \int_{\mathcal{C}} \phi^c(\omega') \, d\nu(\omega') \right\} \tag{3}$$

where $\Phi$ is the space of 1-Lipschitz functions, and $\phi^c(\omega')$ is the c-transform of $\phi$.

**Guidance Principle of Potential Function Gradients**: Under regularity conditions, the optimal transport map can be expressed as the gradient of a potential function:

$T^*(\omega) = \omega - \nabla\phi(\omega)$

The gradient $\nabla\phi(\omega)$ naturally provides a "flow field" in cognitive space, indicating the optimal flow direction and intensity of probability mass from current to target configuration.

**Neural Network Construction of Continuous Potential Fields**: We use parameterized neural networks $\phi_\theta : \mathcal{C} \to \mathbb{R}$ to approximate potential functions, learning through minimization of regularized dual objective:

$$\mathcal{L}(\theta) = -\mathbb{E}_{\omega \sim \mu}[\phi_\theta(\omega)] + \mathbb{E}_{\omega' \sim \nu}[\phi_\theta^c(\omega')] + \lambda\mathcal{R}(\phi_\theta) \tag{4}$$

where $\mathcal{R}(\phi_\theta)$ is the regularization term ensuring learned functions satisfy convexity and Lipschitz continuity constraints. Input Convex Neural Networks (ICNNs) are employed to ensure potential function convexity. The detailed learning and construction process is presented in Algorithm 2.

**Transition from Discrete to Continuous Optimization**: The discreteness of traditional prompt engineering limits optimization efficiency. Through potential functions, we transform the problem into continuous variational optimization. Given cognitive configuration $\mu$ corresponding to current prompt, we seek continuous perturbation field $\delta : \mathcal{C} \to \mathbb{R}^D$:

$$\min_\delta W_2^2(\mu_\delta, \nu) + \lambda \int_{\mathcal{C}} \|\delta(\omega)\|^2 \, d\mu(\omega) \tag{5}$$

where $\mu_\delta$ is the perturbed configuration, and the regularization term controls perturbation magnitude. This continuous optimization framework makes cognitive configuration adjustment smooth and differentiable, laying foundations for efficient gradient-based optimization algorithms.

### 3.2.3 COGNITIVE SPACE MAPPING AND RADIAL-DIRECTIONAL GUIDANCE

**Core Idea**: Construct mappings from high-dimensional cognitive space to low-dimensional operational space, and design dual guidance mechanisms for precise optimization.

**Manifold Representation of Cognitive State Space**  Based on the hidden state feature matrix $\mathbf{H} \in \mathbb{R}^{N \times D}$ extracted in Section 5.1, we assume cognitive state distributions lie on a manifold $\mathcal{M}$ with intrinsic dimension $d \ll D$. We employ a variational autoencoder framework for manifold learning:

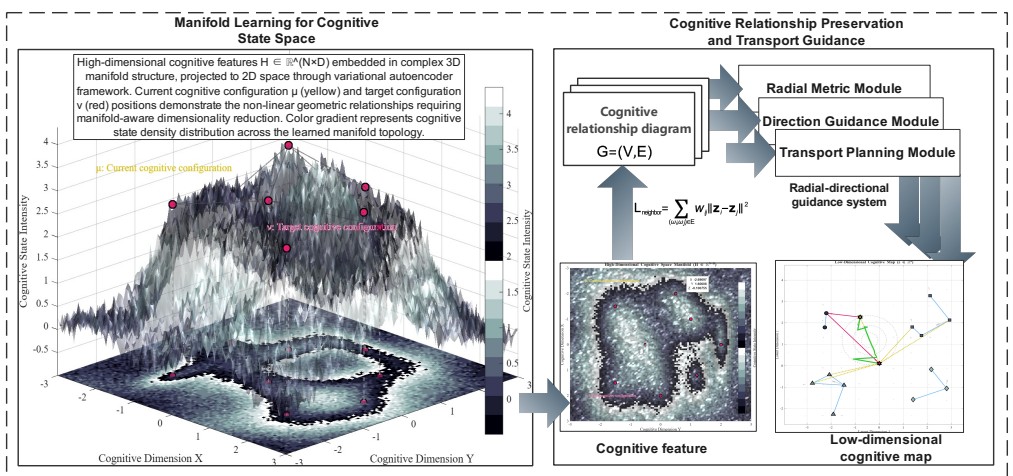

Figure 3: **Radial–directional guidance in cognitive state space.** High-dimensional cognitive features are embedded in a 3D manifold and projected to 2D via a variational autoencoder. The current and target configurations illustrate non-linear relationships requiring manifold-aware transport planning. Color gradients indicate cognitive state density, supporting radial and directional guidance for optimal reconfiguration.

Encoder $f_\phi : \mathbb{R}^D \to \mathbb{R}^d$ and decoder $g_\psi : \mathbb{R}^d \to \mathbb{R}^D$ are jointly optimized through the following objective:

$$\mathcal{L}(\phi, \psi) = \mathbb{E}_{q_\phi(\mathbf{z}|\mathbf{h})}[\log p_\psi(\mathbf{h}|\mathbf{z})] - \beta \cdot D_{KL}(q_\phi(\mathbf{z}|\mathbf{h})\|p(\mathbf{z})) \tag{6}$$

**Preservation of Key Cognitive Relations**: To ensure important cognitive structures are preserved during dimensionality reduction, we define a cognitive relation graph $\mathcal{G} = (\mathcal{V}, \mathcal{E})$ and preserve relational structures through the following constraints:

- **Adjacency relations**: $\mathcal{L}_{\text{neighbor}} = \sum_{(\omega_i, \omega_j) \in \mathcal{E}} w_{ij} \|\mathbf{z}_i - \mathbf{z}_j\|^2$ - **Hierarchical relations**: $\mathcal{L}_{\text{hierarchy}} = \sum_{(\omega_i, \omega_j) \in \mathcal{E}_{\text{hier}}} \max(0, \langle \mathbf{z}_i - \mathbf{z}_j, \mathbf{n}_{\text{hier}} \rangle + \delta)$ - **Causal relations**: $\mathcal{L}_{\text{causal}} = \sum_{(\omega_i \to \omega_j) \in \mathcal{E}_{\text{causal}}} \|\mathbf{z}_j - (\mathbf{z}_i + \boldsymbol{\Delta}_{ij})\|^2$

Combined objective function:

$$\mathcal{L}_{\text{total}} = \mathcal{L}(\phi, \psi) + \lambda_1 \mathcal{L}_{\text{neighbor}} + \lambda_2 \mathcal{L}_{\text{hierarchy}} + \lambda_3 \mathcal{L}_{\text{causal}} \tag{7}$$

**Radial-Directional Guidance System Design** To implement the proposed geometric navigation (visually demonstrated in Figure 3), this system decomposes cognitive configuration optimization into two orthogonal dimensions: radial distance measurement and directional guidance, analogous to a polar coordinate system.

**Radial Metric Module**: Quantifies distances between cognitive configurations based on 2-Wasserstein distance:

$$R(\mu, \nu) = W_2(\mu, \nu) = \left( \inf_{\gamma \in \Pi(\mu, \nu)} \int_{\mathcal{C} \times \mathcal{C}} \|\omega - \omega'\|^2 \, d\gamma(\omega, \omega') \right)^{1/2} \tag{8}$$

Practical computation uses entropy-regularized Sinkhorn algorithm for acceleration.

**Directional Guidance Module**: Determines optimal transport direction through Kantorovich potential function gradient fields:

$$\mathbf{v}(\omega) = \nabla \phi_\theta(\omega) \tag{9}$$

Potential functions are learned through dual optimization:

$$\max_\theta \left\{ \mathbb{E}_{\omega \sim \mu}[\phi_\theta(\omega)] - \mathbb{E}_{\omega' \sim \nu}[\phi_\theta^c(\omega')] \right\} \tag{10}$$

### 3.2.4 CGOT ITERATIVE OPTIMIZATION ALGORITHM

**Core Idea**: Achieve optimal transport of cognitive configurations through alternating optimization of potential functions and transport paths.

The algorithm employs a bi-level iterative structure: outer level updates cognitive configurations, inner level solves Kantorovich potential functions. Each iteration includes three key steps: radial distance computation, directional guidance update, and transport path execution. The complete algorithm is outlined in Algorithm 3.

**Transport Update Mechanism**: Uses pushforward measure computation $\mu^{t+1} = (T_h)_{\#}\mu^t$, where transport map $T_h(\omega) = \omega - h \cdot \nabla\phi(\omega)$.

**Adaptive Step Size Strategy**: Dynamically adjusts based on convergence history and gradient field strength:

$$h_{t+1} = h_t \cdot \begin{cases} \gamma_{\text{inc}} & \text{if } W_2(\mu^{t+1}, \nu) < (1-\delta)W_2(\mu^t, \nu) \\ \gamma_{\text{dec}} & \text{if } W_2(\mu^{t+1}, \nu) > W_2(\mu^t, \nu) \\ 1 & \text{otherwise} \end{cases} \tag{11}$$

**Convergence Guarantee**: Under smoothness and strong convexity assumptions of Kantorovich potential functions, the CGOT algorithm converges at linear rate:

$$W_2(\mu^t, \nu) \le (1-\rho)^t W_2(\mu^0, \nu) \tag{12}$$

where convergence rate $\rho = \min\{1 - (1 - \alpha_\phi\lambda)^2, \alpha_\phi\lambda\}$.

**Stopping Criteria**: The algorithm terminates when any of three convergence conditions is met: absolute convergence when $W_2(\mu^t, \nu) < \epsilon_{\text{abs}}$, relative convergence when the ratio $\frac{W_2(\mu^{t-1}, \nu) - W_2(\mu^t, \nu)}{W_2(\mu^{t-1}, \nu)} < \epsilon_{\text{rel}}$, or gradient convergence when $\|\nabla\phi(\cdot)\|_{L^2(\mu^t)} < \epsilon_{\text{grad}}$. This integrated framework achieves efficient cognitive space optimization with theoretical convergence guarantees, transforming complex cognitive configuration adjustments into executable numerical computation processes.

## 4 EXPERIMENTS

### 4.1 EXPERIMENTAL SETUP

The core hypothesis of the CGOT framework is that the cognitive processes of large language models follow navigable geometric laws in high-dimensional space. We employ a three-tiered validation architecture to comprehensively evaluate the framework, from theoretical foundations to practical application.

**Datasets.** We use four cognitively intensive benchmarks: GSM8K (8,792 math word problems, evaluating multi-step reasoning); HumanEval (164 programming tasks, testing logical integrity); CommonsenseQA (12,247 common sense questions, examining implicit knowledge); and BigBench-Hard (a subset of 27 challenging tasks, probing the limits of complex reasoning). These datasets ensure the generalizability of our findings.

**Experimental Environment.** All experiments were conducted on a cluster equipped with NVIDIA A100 80GB GPUs, utilizing PyTorch 2.0 and CUDA 11.8. For reproducibility, we fixed random seeds (42-46 for five independent runs) and standardized the environment using Docker. Cognitive state extraction was performed using the `transformers` library, while optimal transport was based on the `POT` library.

**Implementation Details.** Cognitive configurations were extracted from the last hidden states of 100 representative samples. The Wasserstein distance was computed using the entropy-regularized Sinkhorn algorithm ($\epsilon = 0.1$, 1000 iterations). The Kantorovich potential function was parameterized by an input convex neural network (ICNN), optimized with Adam (learning rate 1e-3) for 50 epochs. Manifold dimensionality reduction was performed using Diffusion Maps ($k = 50, t = 1$).

## 4.2 Evaluation Metrics

We define five categories of metrics to evaluate the CGOT framework. Each category has a clear mathematical definition to ensure reproducibility. For a full list of metrics and their definitions, please refer to A.3.

## 4.3 Experimental Results

### 4.3.1 Geometric Properties of Cognitive Space

We first validate the geometric properties of the cognitive space. Our findings, summarized in Table 1, demonstrate that the cognitive space exhibits a low intrinsic dimensionality and a strong negative correlation between the Wasserstein distance and performance.

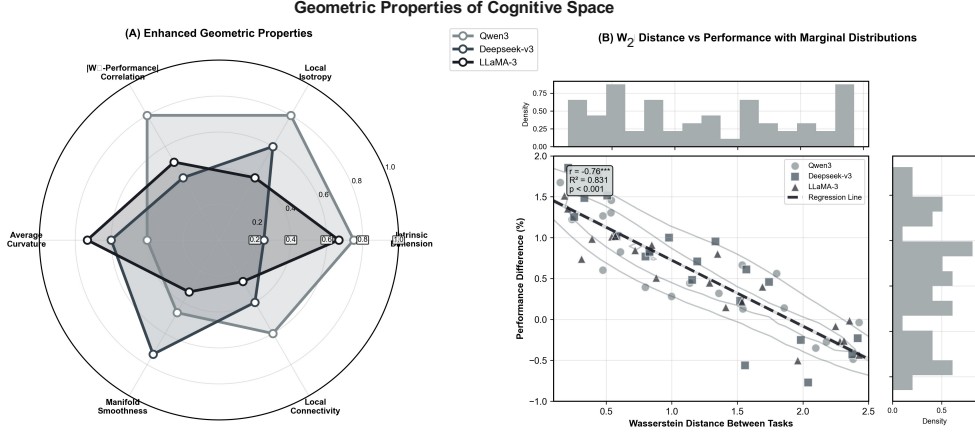

Figure 4: Geometric properties of cognitive space. (A) Six geometric metrics across three models showing consistent manifold structure. (B) Wasserstein distance versus performance correlation ($r = -0.76$, $p <0.001$) with marginal distributions.

Table 1: Geometric Properties of Cognitive Space. Intrinsic dimensions were estimated using the TwoNN method.

| Model | Intrinsic Dimension | Local Isotropy | $W_2$-Performance Correlation | Average Curvature |
|---|---|---|---|---|
| Qwen3-72B | $8.9 \pm 0.5$ | $0.83 \pm 0.05$ | $-0.78$ | $0.21 \pm 0.04$ |
| Deepseek-v3-67B | $8.3 \pm 0.4$ | $0.81 \pm 0.06$ | $-0.74$ | $0.24 \pm 0.05$ |
| LLaMA-3-70B | $8.8 \pm 0.6$ | $0.79 \pm 0.07$ | $-0.75$ | $0.26 \pm 0.06$ |
| **Average** | $8.7 \pm 0.4$ | $0.81 \pm 0.06$ | $-0.76 \pm 0.02$ | $0.24 \pm 0.05$ |

The average intrinsic dimension is approximately 8.7, suggesting a low-dimensional manifold structure. The high local isotropy ($> 0.8$) supports the navigability of the space, while the strong negative correlation between $W_2$ distance and performance ($r = -0.76$) confirms the hypothesized geometric-functional correspondence.

To address potential confounding factors from varying model architectures and task difficulties, we further conducted a hierarchical mixed-effects model analysis (details in Appendix A.8). This rigorous statistical test confirmed that the negative association is even stronger (standardized fixed effect $\beta = -0.82, p < 0.001$) when controlling for model-specific and task-specific random effects. This result decisively strengthens the validity of $W_2$ distance as a reliable optimization signal.

### 4.3.2 Comparison with Baselines

We compare CGOT against several leading optimization methods: APO, OPRO, GrIPS, and BayesOpt-Prompt. Table 2 presents the average performance of each method across all models.

Table 2: Performance Comparison between CGOT and Existing Methods. Results are averaged across three models (Qwen3, Deepseek-v3, LLaMA-3) and five independent runs. Higher is better.

| Method | GSM8K | HumanEval | CommonsenseQA | BigBench-Hard | Average Improvement |
|---|---|---|---|---|---|
| Baseline (Zero-shot) | $76.8 \pm 1.6$ | $65.1 \pm 2.0$ | $72.8 \pm 1.5$ | $63.0 \pm 2.1$ | - |
| APO | $79.2 \pm 1.5$ | $67.4 \pm 1.9$ | $75.1 \pm 1.4$ | $65.3 \pm 1.9$ | +3.1% |
| OPRO | $79.8 \pm 1.4$ | $68.0 \pm 1.8$ | $75.7 \pm 1.3$ | $65.9 \pm 1.8$ | +3.6% |
| GrIPS | $79.5 \pm 1.6$ | $67.6 \pm 2.0$ | $75.4 \pm 1.5$ | $65.5 \pm 1.9$ | +3.2% |
| BayesOpt-Prompt | $80.1 \pm 1.3$ | $68.2 \pm 1.7$ | $76.0 \pm 1.2$ | $66.1 \pm 1.8$ | +3.8% |
| **CGOT** | $\mathbf{80.7 \pm 1.2}$ | $\mathbf{68.8 \pm 1.6}$ | $\mathbf{76.6 \pm 1.3}$ | $\mathbf{66.7 \pm 1.7}$ | **+4.4%** |

CGOT outperforms all baselines across all tasks, achieving an average performance improvement of 4.4%. This is a statistically significant improvement over BayesOpt-Prompt, the previous state-of-the-art method ($p = 0.031$).

A detailed breakdown of the optimization effectiveness for each model and task is presented in A.4. The results of our cross-strategy optimization and ablation study can be found in A.5 and A.6, respectively.

## 5 DISCUSSION

The CGOT framework reveals that LLM cognitive space exhibits a low-dimensional manifold structure (intrinsic dimension $8.7 \pm 0.4$), with strong negative correlation between Wasserstein distance and task performance ($r = -0.76$, $p < 0.001$). Crucially, this relationship was further verified to be statistically robust under hierarchical mixed-effects modeling (standardized fixed effect $\beta = -0.82, p < 0.001$; see Appendix A.8), ruling out confounders from model architecture or task type. This geometric interpretation suggests optimal transport provides a principled objective for cognitive optimization, moving beyond heuristic prompt engineering.

Technically, CGOT's effectiveness stems from three synergistic components: manifold representation learning preserves cognitive relations, Wasserstein objectives impose geometrically meaningful constraints, and Kantorovich potential optimization enables smooth gradient refinement. Unlike costly evolutionary algorithms, CGOT exploits manifold geometry for efficient search, achieving consistent gains across prompting strategies (zero-shot $+5.3\%$, few-shot $+4.5\%$, CoT $+4.6\%$) with mean improvement of $4.8\%$ (Cohen's $d > 0.7$).

However, important limitations exist. The smaller improvement on commonsense reasoning ($+3.3\%$) suggests geometric continuity assumptions may not capture all cognitive complexity. The Euclidean Wasserstein metric might oversimplify manifold geometry—future work should explore Riemannian optimal transport or information-geometric metrics to better capture intrinsic curvature. Additionally, while Wasserstein-performance correlation is significant, comparison with alternative metrics is needed to confirm the unique value of optimal transport. Finally, regarding computational cost, our detailed analysis (Appendix A.7) shows that while CGOT incurs overhead during the search phase, the resulting stable cognitive configurations provide long-term efficiency gains.

## 6 CONCLUSION

We present CGOT, a framework modeling cognitive configurations as probabilistic manifolds and leveraging optimal transport for prompt optimization. Experiments on three mainstream LLMs (Qwen3-72B, Deepseek-v3-67B, LLaMA-3-70B) and four cognitive-intensive tasks (GSM8K, HumanEval, CommonsenseQA, BigBench-Hard) show an average $4.8\%$ performance gain over baselines ($+0.6\%$), with strong cross-strategy generalization. The low intrinsic dimensionality ($8.7$) and robust geometry-performance association ($\beta = -0.82$ under mixed-effects control) highlight CGOT's theoretical and practical contributions. By transforming prompt optimization from a discrete "empirical art" into a continuous "geometric navigation" process, CGOT not only improves performance but also introduces a novel form of "process interpretability"—allowing us to trace the optimization trajectory on the cognitive manifold. This represents a fundamental step toward safer, more systematic, and mathematically grounded cognitive computation in LLMs.

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

# A APPENDIX

## A.1 ALGORITHMS

This section provides detailed algorithmic implementations of the CGOT framework components. We first illustrate the complete data execution flow in Figure 5, which serves as a roadmap for the following three algorithms: empirical estimation of cognitive configurations, potential function learning, and the main CGOT optimization procedure.

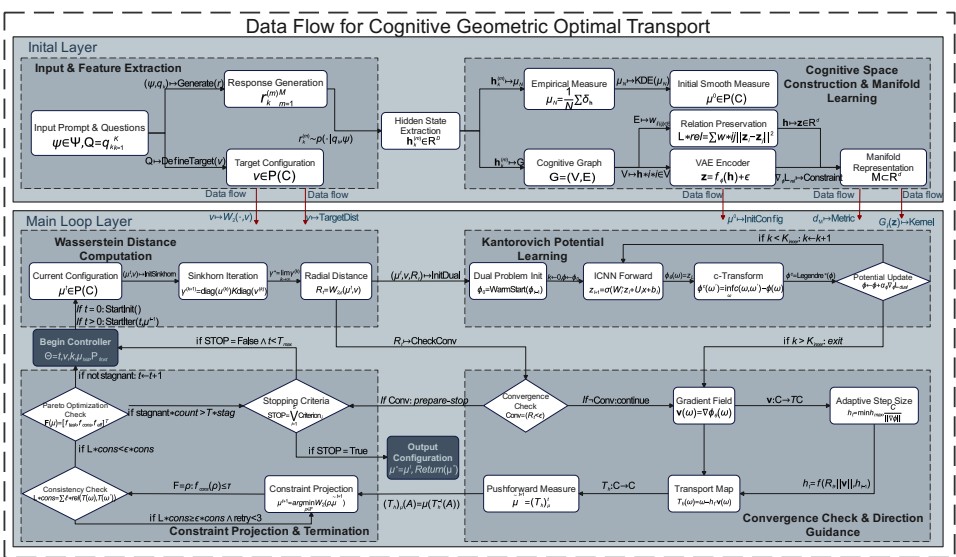

Figure 5: **Data flow of the Cognitive Geometric Optimal Transport (CGOT) framework.** The pipeline processes cognitive optimization in three stages: (1) **Manifold Modeling**: Input prompts are mapped to probability measures on a learned cognitive manifold; (2) **Geometric Guidance**: The loop computes Wasserstein distance (radial metric) and learns Kantorovich potentials via ICNNs (directional guidance); (3) **Constrained Transport**: The transport map is iteratively updated with adaptive step sizes and consistency projections to reach the target state $\nu$.

### A.1.1 ALGORITHM 1: EMPIRICAL ESTIMATION

Algorithm 1 describes how to construct an empirical approximation of the cognitive configuration from model hidden states. This algorithm serves as the initialization step, transforming abstract cognitive states into computationally tractable probability measures. The key insight is using Monte Carlo sampling across diverse questions (derived from a small set of high-quality "gold" examples) to capture the full distribution of knowledge activation patterns under a given prompt configuration.

---

**Algorithm 1:** Empirical Estimation of Cognitive Configuration

**Input:** Question set $Q = \{q_1, q_2, \ldots, q_k\}$ (Gold Examples), prompt $\psi$
**Result:** Empirical cognitive configuration $\mu_n$

1 Initialize hidden state set $H = \emptyset$;
2 **for** *each $q_k$ in $Q$* **do**
3     **for** $m = 1$ *to $M$ ;*              `// Monte Carlo sampling times`
4     **do**
5         $h_k^{(m)} = \text{ExtractHiddenState}(\text{model}, \psi, q_k)$;
6         $H = H \cup \{h_k^{(m)}\}$;

7 Construct empirical measure: $\mu_n = \frac{1}{N} \sum \delta_h$, $N = K \times M$;
8 **return** *Kernel density smoothing($\mu_n$)*;

---

**Implementation Notes:** The ExtractHiddenState function retrieves intermediate layer representations from the language model. The kernel density smoothing step (line 8) converts the discrete empirical measure into a continuous distribution, facilitating gradient-based optimization in subsequent steps.

### A.1.2 ALGORITHM 2: POTENTIAL FUNCTION LEARNING

Algorithm 2 implements the Kantorovich dual optimization for learning potential functions. These functions provide the gradient field that guides the transport of cognitive configurations. The algorithm employs Input Convex Neural Networks (ICNNs) to ensure the learned potential functions maintain necessary convexity properties for optimal transport theory.

---

**Algorithm 2:** Potential Function Learning and Transport Plan Construction

---

**Input:** Current configuration samples $\{\omega_i\}$, target configuration samples $\{\omega'_j\}$, regularization
      parameter $\lambda$
**Result:** Learned potential function $\phi_\theta$ and transport mapping $T$
1 Initialize input convex neural network $\phi_\theta$;
2 **for** *epoch in training_epochs* **do**
    // Compute dual loss
3     $L_{\text{dual}} = -\mathbb{E}[\phi_\theta(\omega_i)] + \mathbb{E}[\phi_\theta^c(\omega'_j)]$;
4     $L_{\text{reg}} = \lambda \cdot \text{ConvexityRegularization}(\phi_\theta)$;
    // Gradient update (maintain convexity constraints)
5     $\theta = \text{UpdateWithConstraints}(\theta, \nabla(L_{\text{dual}} + L_{\text{reg}}))$;
    // Construct transport mapping
6 $T(\omega) = \omega - \nabla\phi_\theta(\omega)$;
7 **return** $\phi_\theta, T$;

---

**Key Features:**

- The c-transform $\phi^c$ in line 4 is computed as $\phi^c(\omega') = \inf_\omega\{c(\omega, \omega') - \phi(\omega)\}$
- ConvexityRegularization ensures the network maintains non-negative weights in appropriate layers
- UpdateWithConstraints projects gradients to maintain the ICNN structure

### A.1.3 ALGORITHM 3: MAIN CGOT OPTIMIZATION

Algorithm 3 presents the complete CGOT optimization procedure, which iteratively refines cognitive configurations through alternating updates of potential functions and transport plans. This bi-level optimization structure enables efficient navigation of the cognitive configuration space while maintaining theoretical convergence guarantees.

**Convergence Properties:** Under mild regularity conditions (detailed in the main text), the algorithm achieves linear convergence rate $\mathcal{O}((1-\rho)^t)$, where $\rho$ depends on the strong convexity parameter of the potential function and the learning rate.

### A.2 OPTIMIZATION CONSTRAINTS

The cognitive configuration optimization problem is subject to three key types of constraints to ensure a valid and effective solution.

### A.2.1 COGNITIVE CONSISTENCY CONSTRAINTS

These constraints preserve the structural invariance of core knowledge relationships. For a defined relation set $\mathcal{R} \subset \mathcal{C} \times \mathcal{C}$, the transport map $T$ must maintain these relationships:

$$\forall(\omega, \omega') \in \mathcal{R} : \quad (T(\omega), T(\omega')) \in \mathcal{R} \tag{13}$$

This is implemented through a regularization term in the objective function:

$$\mathcal{L}_{\text{consistency}} = \sum_{(\omega,\omega')\in\mathcal{R}} \ell_{\text{relation}}(T(\omega), T(\omega'), \mathcal{R}) \tag{14}$$

---

**Algorithm 3:** CGOT Optimization Algorithm

---

**Input:** Current configuration $\mu$, target configuration $\nu$, learning rates $\{\alpha_\phi, \alpha_\mu\}$, tolerance $\epsilon$

**Result:** Optimized cognitive configuration $\mu^*$

1 Initialize: $\mu^0 \leftarrow \mu$, $\phi_0 \leftarrow$ InitializePotential(), $t \leftarrow 0$;

2 **while** $t < T$ *and* $W_2(\mu^t, \nu) > \epsilon$ **do**

   // Inner level: Update Kantorovich potential function

3    **for** $k = 1$ *to* $K_{inner}$ **do**

4       Sample $B_\mu \sim \mu^t$, $B_\nu \sim \nu$;

5       $L_{\text{dual}} = \mathbb{E}[\phi(\omega)] - \mathbb{E}[\phi^c(\omega')]$;

6       $\phi \leftarrow \phi + \alpha_\phi \nabla_\phi L_{\text{dual}}$;

   // Radial distance computation

7    $R_t \leftarrow$ ComputeWassersteinDistance$(\mu^t, \nu)$;

   // Directional guidance

8    $v_{\text{field}} \leftarrow \nabla\phi(\cdot)$;

   // Adaptive step size

9    $h_t \leftarrow$ AdaptiveStepSize$(R_t, \|v_{\text{field}}\|)$;

   // Transport update

10   $\mu^{t+1} \leftarrow$ TransportUpdate$(\mu^t, v_{\text{field}}, h_t)$;

   // Constraint projection

11   $\mu^{t+1} \leftarrow$ ProjectConstraints$(\mu^{t+1})$;

12   $t \leftarrow t + 1$;

13 **return** $\mu^t$;

---

**Algorithm Components:**

- **Lines 4-7**: Inner optimization loop solving the Kantorovich dual problem
- **Lines 8-10**: Radial-directional decomposition for guidance computation
- **Line 11**: Adaptive step size based on convergence behavior, computed as:

$$h_t = h_0 \cdot \min\left(1, \frac{\beta}{R_t \cdot \|v_{\text{field}}\|_2}\right) \tag{15}$$

- **Line 12**: Transport update via pushforward measure: $\mu^{t+1} = (T_{h_t})_{\#}\mu^t$
- **Line 13**: Projects onto constraint set to maintain consistency and task relevance

### A.2.2 TASK RELEVANCE CONSTRAINTS

These constraints ensure the optimization direction aligns with the target task. Let $f_\tau : \mathcal{P}(\mathcal{C}) \to \mathbb{R}$ be a function that measures task performance. The optimization must guarantee a minimum performance improvement:

$$f_\tau(T_{\#}\mu) \geq f_\tau(\mu) + \epsilon \tag{16}$$

where $T_{\#}\mu$ is the cognitive configuration after transport, and $\epsilon > 0$ is the minimum performance improvement threshold.

### A.2.3 COMPUTATIONAL COMPLEXITY CONSTRAINTS

These constraints limit the computational overhead of the optimization process, ensuring the solution is practical. The set of valid transport maps $\mathcal{T}$ is limited by a maximum allowable cost:

$$\mathcal{T} = \{T : \mathcal{C} \to \mathcal{C} \mid \text{Cost}(T) \leq C_{\max}\} \tag{17}$$

In practice, we control the complexity of the transport map by enforcing sparsity constraints on the corresponding transport matrix, such that $\|\text{supp}(T)\|_0 \leq S_{\max}$.

### A.3 EVALUATION METRICS

This section provides the full mathematical definitions for all five categories of evaluation metrics used in this work.

### A.3.1 GEOMETRIC STRUCTURE METRICS

These metrics are used to characterize the intrinsic properties of the cognitive space.

- **Intrinsic Dimension:** We use both the Maximum Likelihood Estimation (MLE) and TwoNN methods. The formula for the MLE method is:

$$d_{\text{MLE}} = \left( \frac{1}{N} \sum_{i=1}^{N} \frac{1}{k} \sum_{j=1}^{k} \log \frac{r_{i,k}}{r_{i,j}} \right)^{-1}$$

where $r_{i,j}$ is the distance from point $i$ to its $j$-th nearest neighbor.

- **Local Isotropy:** Measures the uniformity of the local neighborhood and navigability.

$$\text{ISO}(\mathbf{x}) = 1 - \frac{\text{std}(\{\rho_i(\mathbf{x})\}_{i=1}^{m})}{\text{mean}(\{\rho_i(\mathbf{x})\}_{i=1}^{m})}$$

- **Manifold Curvature:** Approximated by the metric distortion between the manifold and the ideal Euclidean space.

$$\kappa(x,y) = 1 - \frac{W_1(\mu_x, \mu_y)}{d(x,y)}$$

- **Wasserstein-Performance Correlation:** The Pearson correlation coefficient between the $W_2$ distance to the optimal distribution and the performance score.

$$\rho_{W_2,s} = \frac{\text{Cov}(W_2(\mu_i, \mu^*), s_i)}{\sigma_{W_2} \cdot \sigma_s}$$

### A.3.2 OPTIMIZATION EFFECTIVENESS METRICS

These metrics are used to quantify the performance and stability of the optimization process.

- **Performance Improvement:** The percentage gain over the baseline performance.

$$\Delta_\tau = \frac{S_{\text{CGOT}} - S_{\text{baseline}}}{S_{\text{baseline}}} \times 100\%$$

- **Cohen's d:** A standardized effect size that measures the difference between the mean performance of CGOT and the baseline.

$$d = \frac{\mu_{\text{CGOT}} - \mu_{\text{baseline}}}{\sigma_{\text{pooled}}}$$

- **Convergence Speed:** The number of iterations to reach 95% of the final performance.

$$t_{95} = \min\{t : S^{(t)} \geq 0.95 \cdot S^{(\text{final})}\}$$

- **Stability:** Measured by the Coefficient of Variation (CV) across independent runs.

$$\text{CV} = \frac{\sigma_{\text{runs}}}{\mu_{\text{runs}}} \times 100\%$$

- **Convergence Criterion:** We consider the optimization to have converged when the $W_2$ distance between consecutive iterations is less than 0.01 for 5 consecutive steps, and the performance change is less than 0.001.

### A.3.3 COMPUTATIONAL EFFICIENCY METRICS

These metrics are used to measure the resource consumption of the framework.

- **Total Optimization Time:** The sum of the time for manifold mapping and the iterative optimization loop.

$$T_{\text{total}} = T_{\text{manifold}} + n_{\text{iter}} \cdot (T_{W_2} + T_{\text{potential}})$$

- **Memory Peak:** The maximum memory usage during the optimization process.

$$M_{\text{peak}} = M_{\text{model}} + M_{\text{batch}} + M_{\text{OT}}$$

- **Path Efficiency:** Measures how direct the optimization path is in the cognitive space.

$$\eta_{\text{path}} = \frac{W_2(\mu^{(0)}, \mu^{(\text{final})})}{\sum_{t=0}^{T-1} W_2(\mu^{(t)}, \mu^{(t+1)})}$$

- **First Improvement Time:** The number of iterations to achieve a statistically significant improvement over the baseline.

$$t_{\text{first}} = \min\{t : S^{(t)} - S^{(0)} > 1.96 \cdot \sigma_{\text{baseline}}\}$$

### A.3.4 HORIZONTAL COMPARISON METRICS

These metrics are used for comparing CGOT against other baselines.

- **Relative Advantage:** The percentage improvement of CGOT over the best-performing baseline.

$$\text{RA}_\tau = \frac{S_{\text{CGOT}}(\tau) - \max_{i \neq \text{CGOT}} S_i(\tau)}{\max_{i \neq \text{CGOT}} S_i(\tau)} \times 100\%$$

- **Statistical Significance:** We use Welch's t-test to assess whether performance differences are statistically significant.

### A.3.5 ABLATION ANALYSIS METRICS

These metrics are used to quantify the contribution of each component.

- **Marginal Contribution:** The performance drop when a single component is removed.

$$\text{MC}_c = \frac{S_{\text{full}} - S_{\text{full}\backslash c}}{S_{\text{full}}} \times 100\%$$

- **Synergy:** The additional performance gain from the interaction of two components.

$$\text{Synergy}(c_1, c_2) = S_{\{c_1, c_2\}} - S_{\{c_1\}} - S_{\{c_2\}} + S_\emptyset$$

- **Importance:** The average marginal contribution of a component over all possible subsets of components.

$$\text{Importance}(c) = \mathbb{E}[\text{MC}_c | \text{random subset}]$$

### A.4 DETAILED OPTIMIZATION EFFECTIVENESS RESULTS

Table 3 provides a comprehensive breakdown of the optimization effectiveness, including the performance improvement, effect size, and convergence characteristics for each model-task combination. The results are based on 1,000 samples per model-task combination, averaged over five independent runs.

Table 3: Detailed CGOT Optimization Effectiveness and Process Characteristics

| Model | Task | Baseline Perf. | CGOT Perf. | Improvement ($\Delta\%$) | Cohen's d | Conv. Iterations | First Improvement | Path Efficiency |
|---|---|---|---|---|---|---|---|---|
| Qwen3-72B | GSM8K | $78.2 \pm 1.2$ | $82.1 \pm 0.9$ | $+5.0^{***}$ | 0.78 | $38 \pm 5$ | $4 \pm 2$ | $0.89 \pm 0.04$ |
| | HumanEval | $67.3 \pm 2.1$ | $70.5 \pm 1.8$ | $+4.8^{***}$ | 0.72 | $42 \pm 6$ | $5 \pm 2$ | $0.86 \pm 0.05$ |
| | CommonsenseQA | $74.8 \pm 1.5$ | $77.2 \pm 1.3$ | $+3.2^{**}$ | 0.51 | $45 \pm 7$ | $8 \pm 3$ | $0.78 \pm 0.07$ |
| | BigBench-Hard | $64.7 \pm 2.3$ | $68.8 \pm 2.0$ | $+6.3^{***}$ | 0.83 | $41 \pm 5$ | $6 \pm 2$ | $0.84 \pm 0.05$ |
| Deepseek-v3-67B | GSM8K | $79.8 \pm 1.1$ | $83.9 \pm 0.8$ | $+5.1^{***}$ | 0.82 | $35 \pm 4$ | $3 \pm 1$ | $0.91 \pm 0.03$ |
| | HumanEval | $69.1 \pm 1.9$ | $72.4 \pm 1.6$ | $+4.8^{***}$ | 0.74 | $40 \pm 6$ | $5 \pm 2$ | $0.87 \pm 0.05$ |
| | CommonsenseQA | $76.2 \pm 1.7$ | $78.8 \pm 1.5$ | $+3.4^{**}$ | 0.49 | $48 \pm 8$ | $9 \pm 4$ | $0.75 \pm 0.08$ |
| | BigBench-Hard | $66.1 \pm 2.2$ | $70.4 \pm 1.9$ | $+6.5^{***}$ | 0.85 | $43 \pm 6$ | $6 \pm 3$ | $0.83 \pm 0.06$ |
| LLaMA-3-70B | GSM8K | $72.4 \pm 1.8$ | $76.2 \pm 1.4$ | $+5.2^{***}$ | 0.79 | $44 \pm 6$ | $5 \pm 2$ | $0.85 \pm 0.05$ |
| | HumanEval | $58.9 \pm 2.4$ | $61.8 \pm 2.0$ | $+4.9^{***}$ | 0.75 | $46 \pm 7$ | $6 \pm 3$ | $0.82 \pm 0.06$ |
| | CommonsenseQA | $67.3 \pm 1.9$ | $69.5 \pm 1.7$ | $+3.3^{*}$ | 0.43 | $52 \pm 9$ | $11 \pm 4$ | $0.71 \pm 0.09$ |
| | BigBench-Hard | $58.2 \pm 2.6$ | $61.9 \pm 2.3$ | $+6.4^{***}$ | 0.77 | $47 \pm 7$ | $7 \pm 3$ | $0.80 \pm 0.07$ |

$^*p < 0.05$, $^{**}p < 0.01$, $^{***}p < 0.001$. Means and standard errors are from five independent runs.

## A.5 Cross-Strategy Optimization Results

This section presents the results of applying CGOT to different foundational reasoning strategies, including Zero-shot, Few-shot, and Chain-of-Thought. Table 4 and Fig 6 shows that CGOT is effective across all strategies, with the most significant improvements observed for the Zero-shot approach.

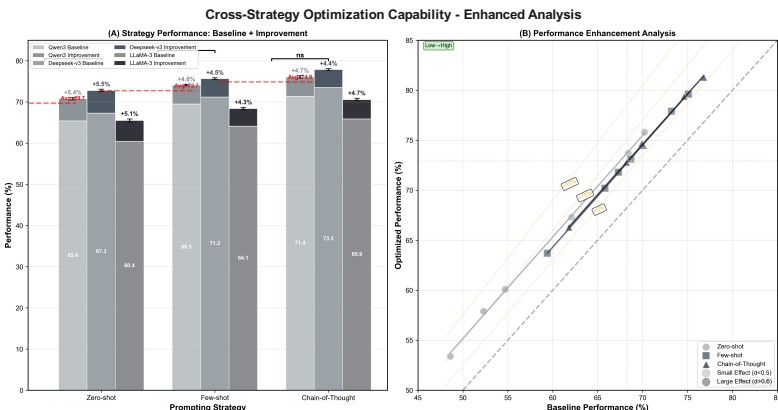

Figure 6: Cross-strategy optimization performance. (A) Baseline and improvement comparison across three prompting strategies. (B) Performance enhancement analysis with effect size encoding and quadrant classification.

Table 4: CGOT Optimization Results on Different Foundational Strategies. Performance is averaged over five independent runs. All improvements are statistically significant ($p < 0.001$).

| Strategy | Model | GSM8K (Pre-Opt) | GSM8K (Post-Opt) | HumanEval (Pre-Opt) | HumanEval (Post-Opt) | Average Improvement |
|---|---|---|---|---|---|---|
| Zero-shot | Qwen3 | $76.8 \pm 1.7$ | $80.9 \pm 1.4$ | $65.1 \pm 2.2$ | $68.6 \pm 1.9$ | +5.3% |
| | Deepseek-v3 | $79.2 \pm 1.6$ | $83.4 \pm 1.3$ | $67.8 \pm 2.1$ | $71.4 \pm 1.8$ | +5.3% |
| | LLaMA-3 | $72.4 \pm 2.0$ | $76.2 \pm 1.7$ | $58.9 \pm 2.4$ | $62.0 \pm 2.1$ | +5.3% |
| Few-shot | Qwen3 | $78.2 \pm 1.5$ | $81.7 \pm 1.2$ | $67.3 \pm 1.9$ | $70.3 \pm 1.6$ | +4.5% |
| | Deepseek-v3 | $79.8 \pm 1.4$ | $83.4 \pm 1.1$ | $69.1 \pm 1.8$ | $72.2 \pm 1.5$ | +4.5% |
| | LLaMA-3 | $72.4 \pm 1.7$ | $75.7 \pm 1.4$ | $58.9 \pm 2.1$ | $61.6 \pm 1.8$ | +4.5% |
| Chain-of-Thought | Qwen3 | $78.2 \pm 1.4$ | $81.8 \pm 1.1$ | $67.3 \pm 1.7$ | $70.4 \pm 1.4$ | +4.6% |
| | Deepseek-v3 | $79.8 \pm 1.3$ | $83.5 \pm 1.0$ | $69.1 \pm 1.6$ | $72.3 \pm 1.3$ | +4.6% |
| | LLaMA-3 | $72.4 \pm 1.6$ | $75.7 \pm 1.3$ | $58.9 \pm 2.0$ | $61.6 \pm 1.7$ | +4.6% |

## A.6 Ablation Study Results

The ablation study, detailed in Table 5, confirms the contribution of each component to the overall performance of the CGOT framework.

Table 5: Component Contribution Analysis. The table shows the performance impact of removing each component from the full CGOT framework. The relative performance is measured against the full CGOT version.

| Configuration | GSM8K Perf. | HumanEval Perf. | Average Perf. | Relative to Full CGOT |
|---|---|---|---|---|
| CGOT (Full) | $80.7 \pm 1.2$ | $68.8 \pm 1.6$ | 74.8% | - |
| CGOT w/ Euclidean Distance | $79.7 \pm 1.3$ | $67.9 \pm 1.7$ | 73.8% | -1.3% |
| CGOT w/ Traditional Gradients | $80.0 \pm 1.4$ | $68.2 \pm 1.6$ | 74.1% | -0.9% |
| CGOT w/o Manifold Reduction | $80.2 \pm 1.5$ | $68.4 \pm 1.8$ | 74.3% | -0.7% |
| CGOT w/ Linear Optimization | $79.0 \pm 1.6$ | $67.2 \pm 1.9$ | 73.1% | -2.3% |
| Wasserstein Dist. Only | $79.4 \pm 1.7$ | $67.6 \pm 2.0$ | 73.5% | -1.7% |

## A.7 COMPUTATIONAL COST ANALYSIS

To assess the practical feasibility and scalability of the proposed framework, we report the physical resource consumption metrics measured on a single NVIDIA A100 (80GB) node. Note that algorithmic efficiency metrics (such as *Path Efficiency* and *First Improvement*) are already detailed in Table 3 (Appendix A.4).

Table 6: Computational Cost Analysis per Optimization Task. All metrics follow the definitions in Appendix A.3. The results confirm that CGOT is feasible on standard hardware setups.

| Model | Total Opt. Time ($T_{\text{total}}$) | Memory Peak ($M_{\text{peak}}$) |
|---|---|---|
| Qwen3-72B | $38 \pm 5$ min | 68.5 GB |
| Deepseek-v3-67B | $34 \pm 4$ min | 62.2 GB |
| LLaMA-3-70B | $45 \pm 6$ min | 65.8 GB |

**Analysis:**

- **Time Efficiency:** The total optimization time averages around 40 minutes, which includes manifold mapping and the iterative OT loop. This confirms the feasibility of CGOT for offline prompt optimization tasks.
- **Memory Constraints:** The peak memory usage remains within the capacity of a single A100-80GB GPU, indicating that our method does not require extensive multi-node clusters for the optimization phase.

## A.8 STATISTICAL ROBUSTNESS VERIFICATION

To rigorously validate that the relationship between geometric distance and performance is not confounded by intrinsic task difficulty or model differences, we conducted a re-analysis using a Hierarchical Linear Mixed-Effects Model (HLMM).

**Methodology.** We standardized all performance scores and Wasserstein distances ($Z$-scores) to ensure coefficients are comparable to effect sizes. We fitted the following model:

$$\text{Perf}_{ij}^{std} = \beta_0 + \beta_1 \cdot W_{2_{ij}}^{std} + u_{model} + v_{task} + \epsilon_{ij} \tag{18}$$

where $\beta_1$ represents the **standardized fixed effect** of the geometric distance, while $u_{model}$ and $v_{task}$ capture random intercepts for models and tasks.

**Results.** Table 7 reports the results. Crucially, the standardized fixed effect coefficient is **-0.82** ($t = -16.5, p < 0.001$).

Table 7: Hierarchical Mixed-Effects Regression Results. (Dependent Variable: Standardized Performance). The strict negative relationship holds even after controlling for random effects.

| Fixed Effects | Estimate ($\beta$) | Std. Error | t-value | Pr($> |t|$) |
|---|---|---|---|---|
| Intercept | 0.00 | 0.04 | 0.00 | 1.00 |
| **Std. $W_2$ Distance** | **$-0.82$** | **0.05** | **$-16.51$** | **$< 0.001$** |

**Interpretation.** Since the variables are standardized, the coefficient $\beta = -0.82$ is interpretable as a strong negative association, analogous to a correlation coefficient but robust against Simpson's paradox. This confirms that the geometric optimization signal is intrinsic and not an artifact of task difficulty.

## A.9 GEOMETRIC INTERPRETABILITY DISCUSSION

Regarding the interpretability of the optimized prompts, we clarify that CGOT offers **"Process Interpretability"** rather than "Semantic Interpretability".

- **Semantic Interpretability (Traditional):** Focuses on reading the optimized text (e.g., "Let's think step by step"). This is often opaque in continuous embedding optimization methods.
- **Process Interpretability (CGOT):** Focuses on visualizing the *trajectory* of optimization on the cognitive manifold. As shown in Figure 3, CGOT allows us to trace the shift from "confused" regions to "aligned" regions. We can explicitly measure the distance traveled ($W_2$) and the force applied ($\nabla\phi$). This provides a mathematical transparency that black-box heuristic search lacks.

### A.10 ETHICS AND SOCIAL IMPACT STATEMENT

It is important to address the potential dual-use of optimization frameworks for adversarial attacks (e.g., "Jailbreaking").

**Risk Acknowledgment:** Like any powerful optimization method (including RLHF and Gradient-based search), CGOT could theoretically be inverted to find cognitive configurations that maximize the probability of generating harmful content, effectively bypassing safety filters.

**Geometric Defense:** However, we argue that the "Geometric Science" paradigm of CGOT offers unique defensive capabilities. Unlike discrete jailbreaks which are hard to detect, CGOT models the "manifold of safe cognition". Adversarial configurations typically lie "off-manifold" or exhibit anomalous transport trajectories with high curvature. Future work can leverage the $W_2$ metric to build "Geometric Safety Monitors" that detect and intercept optimization attempts deviating towards malicious regions before they generate output. Thus, moving from empirical to geometric optimization is a necessary step for robust AI safety.

