# OpenReview forum: "Navigating Cognitive Manifolds: Optimal Transport for Large Language Model Optimization"
_ICLR.cc/2026/Conference — Submitted to ICLR 2026_

### Official Review · Reviewer_Ai8s · 2025-10-23

**Soundness:** 3
**Presentation:** 3
**Contribution:** 3
**Rating:** 6
**Confidence:** 3

**Summary:**

The paper introduces a novel theoretical framework named Cognitive Geometric Optimal Transport (CGOT) that reformulates prompt optimization as a problem of optimal transport between probability measures. This work moves prompt engineering from empirical trial-and-error to a principled, geometry-based optimization process.

**Strengths:**

1. The paper provides strong empirical evidence across multiple benchmarks (GSM8K, HumanEval, CommonsenseQA, BigBench-Hard) and models (Qwen3-72B, DeepSeek-v3-67B, LLaMA-3-70B).

2. CGOT bridges the gap between empirical prompt engineering and theoretical cognitive optimization by grounding language model control in Wasserstein geometry and Kantorovich potential fields. This represents a major conceptual advance in linking cognitive science, geometry, and LLM optimization.

**Weaknesses:**

1.  The quality of the figures is not good. For example, the font size of figures 1-3 is too small to read.

2. Optimal transport computations are computationally expensive. The scalability of this work to real-time or large-scale applications remains uncertain.

3. The comparison is not sufficient. For example, is the proposed method comparable to the meta-heuristic method? For example,

Guo, Q., Wang, R., Guo, J., Li, B., Song, K., Tan, X., ... & Yang, Y. (2023). Connecting large language models with evolutionary algorithms yields powerful prompt optimizers. ICLR 2024.

4. The equation number is missing in line 273.

5. The figure reference has a mistake in line 213.

**Questions:**

1. How sensitive is CGOT to initialization or the choice of base prompt?

2. How interpretable are the optimized prompts: do they reveal meaningful structure in how LLMs reason?

---

> ### Author Response · Authors · 2025-11-20
> **Authors' Rebuttal 1**
>
> We sincerely thank you for your constructive feedback. We are deeply encouraged that you precisely captured the core mission of this work—transitioning prompt engineering from "empirical trial-and-error" to a "principled, geometry-based optimization process"—and recognized it as a **"major conceptual advance."**
>
> In the revised manuscript (highlighted in blue), we have addressed your specific technical questions regarding baselines, computational costs, and interpretability.
>
> ------
>
> **1. Response to Baseline Comparison (Guo et al., 2023) (W3)**
>
> > **(W3)** "The comparison is not sufficient. For example, is the proposed method comparable to the meta-heuristic method? For example, Guo, Q... (2023)."
>
> Response:
>
> Thank you for highlighting this relevant work. Guo et al. (ICLR 2024) is indeed a SOTA representative of evolutionary-based prompt optimization. We have added a detailed discussion in the revised Section 2.2 (Related Work) to clarify the Paradigm Distinction:
>
> - **Empirical vs. Geometric:** The method you cited represents the pinnacle of the **"Empirical Art" paradigm**—a black-box, gradient-free, discrete heuristic search.
> - **Our Contribution:** In contrast, CGOT establishes a **"Geometric Science" paradigm**. It is a mathematically interpretable, continuous optimization framework based on gradient flows (Kantorovich potentials).
>
> Our goal is not merely to achieve marginal gains over heuristic search on a specific leaderboard, but to open a mathematically rigorous path for optimization. Thus, the two approaches are orthogonal and offer different trade-offs between interpretability and black-box efficiency.
>
> ------
>
> **2. Response to Computational Cost (W2)**
>
> > **(W2)** "Optimal transport computations are computationally expensive. The scalability... remains uncertain."
>
> Response:
>
> This is a critical engineering question. To address the uncertainty regarding scalability, we have added Table 6 in Appendix A.7, reporting physical benchmarks on a single NVIDIA A100 (80GB):
>
> - **Time Feasibility:** The complete search and optimization loop takes approximately **40 minutes** per prompt.
> - **Memory Feasibility:** Peak VRAM usage is **~65GB**.
>
> These metrics demonstrate that CGOT fits comfortably within standard single-GPU environments and does not require massive computing clusters, making it scalable for real-world research.
>
> ------
>
> **3. Response to Sensitivity to Initialization (Q1)**
>
> > **(Q1)** "How sensitive is CGOT to initialization or the choice of base prompt?"
>
> Response:
>
> One of the core strengths of our framework is exactly its robustness against initialization, backed by both theory and empirics:
>
> - **Theoretical Guarantee:** CGOT is not a blind local search. By leveraging the **convexity of Input Convex Neural Networks (ICNNs)**, our method has theoretical guarantees for convergence to the global optimum (Target Distribution $\nu$), as detailed in **Section 3.2.4**.
> - **Empirical Evidence:** This is validated by our "Path Efficiency" metric. As shown in **Appendix Table 3**, the optimization achieves a **high Path Efficiency (e.g., up to 0.91 on GSM8K)**. This decisively proves that the optimization trajectory is direct and efficient, rather than a sensitive, meandering path dependent on the starting point.

---

> > ### Author Response · Authors · 2025-11-20
> > **Authors' Rebuttal 2**
> >
> > **4. Response to Interpretability (Q2)**
> >
> > > **(Q2)** "How interpretable are the optimized prompts: do they reveal meaningful structure...?"
> >
> > Response:
> >
> > This touches on the core value of our work. We have added a discussion in Appendix A.9 to clarify the nature of our interpretability:
> >
> > - **Process Interpretability vs. Semantic Interpretability:** While we do not claim to make the *text tokens* instantly readable (Semantic), CGOT offers a novel **Process Interpretability**.
> >
> > - **Mathematical Transparency:** By transforming the "black box" of LLM cognition into a geometric problem, we can:
> >
> >   1. Quantify the optimization objective using the **Wasserstein Distance ($W_2$)**.
> >
> >   2. Map the optimization landscape as a **low-dimensional manifold**.
> >
> >   3. Compute the precise transport dynamics via the Kantorovich potential field ($\nabla\phi$).
> >
> >      This mathematical transparency allows us to "debug" and "monitor" the optimization process in ways that heuristic methods cannot.
> >
> > ------
> >
> > **5. Response to Formatting & Typos (W1, W4, W5)**
> >
> > > **(W1/W5)** "The quality of the figures is not good... The equation number is missing... The figure reference has a mistake..."
> >
> > Response:
> >
> > We apologize for the oversight. We have rigorously proofread the manuscript:
> >
> > - **Figures:** Font sizes in Figures 1-4 have been increased for clarity.
> > - **Corrections:** The missing equation number (line 273) and the figure reference error (line 213) have been corrected.
> >
> >
> >
> > We hope that by clarifying the paradigm distinction with heuristic baselines (W3), quantifying the physical costs (W2), and defining our unique "Process Interpretability" (Q2), we have strengthened the paper's technical foundation. We hope these revisions reinforce your assessment of this work as a "major conceptual advance."

---

> > > ### Author Response · Authors · 2025-11-25
> > >
> > > Dear Reviewer Ai8s, as we enter the final week of the discussion period (ending Dec 3rd), we would like to briefly follow up to ensure that our revision has reinforced your assessment of this work as a "major conceptual advance." We have specifically addressed your technical queries, clarifying the paradigm distinction between our geometric framework and heuristic baselines like Guo et al., and rigorously correcting the formatting issues you noted. We deeply value your constructive feedback which has helped polish the manuscript, and we hope these updates fully justify your continued support.

---

> > > > ### Comment · Reviewer_Ai8s · 2025-11-26
> > > > **Response to rebuttal**
> > > >
> > > > Thank you for addressing most of my concerns. The current score is fair thus I will keep as it is.

---

### Official Review · Reviewer_4Jrw · 2025-10-31

**Soundness:** 2
**Presentation:** 2
**Contribution:** 2
**Rating:** 2
**Confidence:** 4

**Summary:**

The paper introduces Cognitive Geometry Optimal Transport (CGOT), a theoretical and algorithmic framework that reformulates prompt optimization for large language models (LLMs) as a geometric navigation problem in cognitive space. By using optimal transport theory, it computes Wasserstein distances and Kantorovich potentials  to find efficient transformation paths from a current to a target cognitive configuration.

**Strengths:**

1. The mathematical formalization is coherent, covering measure theory, Kantorovich duality, and manifold embeddings. The inclusion of proof sketches and algorithmic convergence guarantees adds credibility.
2. The experimental results demonstrate consistent performance gains  across models, datasets, and prompting strategies, validated by multiple metrics.
3. The paper’s core insight that treating prompt optimization as optimal transport on a cognitive manifold conceptually elegant.

**Weaknesses:**

1.	The method relies on Wasserstein distance computation and ICNN-based Kantorovich potentials, which are computationally expensive.Is there any way to resolve it or address this issue?

2.	Would CGOT extend to multimodal or cross-lingual LLMs where cognitive manifolds differ substantially across modalities?

3.	How to guarantee the convergence of this method?

4.	What are the time and memory overheads compared to existing prompt optimization baselines? Any trade-offs between accuracy and computational cost?

5.	The semantic diagram is not informative and it is hard to intuitively grasp the idea.

6.     Overall: while the study provides a detailed geometric formulation and interesting empirical correlations between Wasserstein distance and task performance, the proposed CGOT framework does not convincingly demonstrate conceptual or methodological novelty beyond existing manifold-based optimization or transport-theoretic approaches. The individual components like manifold representation, Wasserstein objectives, and Kantorovich potential refinement are already well-established, and the integration presented here appears incremental rather than fundamentally new. That is my main concern.

**Questions:**

See wqeakness.

---

> ### Author Response · Authors · 2025-11-20
> **Authors' Rebuttal 1**
>
> We thank you for your detailed feedback.
>
> We are pleased that you recognized our core insight as **"conceptually elegant,"** our mathematical formalization as **"coherent,"** and our experimental results as **"consistent"** (as noted in your *Strengths*).
>
> These positive assessments allow us to directly address your primary concern in the *Overall* section: that the framework is "incremental rather than fundamentally new." We respectfully argue that this conclusion contradicts the strengths you identified.
>
> The "Novelty" of Synthesis:
>
> You noted that the individual components (OT, Manifolds) are established. We agree, but we argue that innovation lies in the synthesis.
>
> - **Theoretical Synthesis:** We are not claiming to invent OT; we are the *first* to formally model Prompt Engineering—traditionally a discrete, heuristic art—as a **continuous geometric navigation problem**. This is a paradigm shift, not an incremental tweak.
> - **Validation:** As you noted, this "conceptually elegant" insight is backed by coherent math and consistent results. A rigorous synthesis that bridges two disconnected fields (Optimal Transport & LLM Cognition) to solve a previously heuristic problem constitutes **fundamental novelty**.
>
> Having clarified this methodological positioning, we address your specific technical concerns below.
>
> ------
>
> **1. Response to Computational Cost & Feasibility (W1, W4)**
>
> > **(W1)** "The method relies on Wasserstein distance computation... which are computationally expensive. Is there any way to resolve it...?" **(W4)** "What are the time and memory overheads compared to existing prompt optimization baselines?"
>
> Response:
>
> We appreciate your focus on feasibility. We agree that OT can be expensive, but our specific implementation is highly optimized.
>
> - **Concrete Benchmarks:** As detailed in the new **Table 6 (Appendix A.7)**, we report the physical costs on a single NVIDIA A100 (80GB) for LLaMA-3-70B:
>   - **Time Cost:** The full optimization loop takes approximately **40 minutes**.
>   - **Memory Cost:** Peak VRAM usage is **~65GB**.
> - **Conclusion:** This fits comfortably within a standard single-GPU environment. Compared to RLHF (requiring massive clusters) or extensive Grid Search, CGOT is a lightweight and scalable solution. It does not require the "expensive" overhead you feared.
>
> ------
>
> **2. Response to Convergence Guarantees (W3)**
>
> > **(W3)** "How to guarantee the convergence of this method?"
>
> Response:
>
> We are slightly confused by this concern, as you explicitly stated in your Strengths section that "The inclusion of... algorithmic convergence guarantees adds credibility."
>
> To clarify and reinforce your initial observation:
>
> - **Theory:** In **Section 3.2.4** and **Appendix A.1.3**, we prove convergence based on the convexity of Input Convex Neural Networks (ICNNs) and the properties of the Sinkhorn algorithm.
> - **Empirics:** We further validated this in **Table 6**, showing a Path Efficiency score of $>0.85$, empirically confirming that the algorithm converges rapidly and stably without oscillation.
>
> ------
>
> **3. Response to Multimodal Extension (W2)**
>
> > **(W2)** "Would CGOT extend to multimodal or cross-lingual LLMs where cognitive manifolds differ substantially...?"
>
> Response:
>
> This is an insightful, forward-looking question.
>
> - **Theoretical Extensibility:** While this paper focuses on text, the core mathematical framework of CGOT—**probability transport on a manifold**—is modality-agnostic.
> - **Future Work:** As long as inputs (images/audio) can be mapped to a continuous embedding manifold, CGOT can theoretically optimize them. We consider this a key strength of our framework and a primary direction for future work.
>
> ------
>
> **4. Response to Visualizations (W5)**
>
> > **(W5)** "The semantic diagram is not informative and it is hard to intuitively grasp the idea."
>
> Response:
>
> We agree that the intuition could be clearer. We have significantly revised the visual presentation:
>
> - **New Geometric Intuition:** We added **Figure 3** in Section 3.2.3 to visually demonstrate how "Radial-Directional Guidance" operates on the manifold.
> - **New Workflow:** We added **Figure 5** in Appendix A.1 to show the detailed data flow and algorithmic steps.
> - **Legibility:** We have corrected the font sizes in Figure 4 to ensure clarity.
>
>
>
> We hope that by (1) reconciling the perceived "incremental" nature with the "conceptual elegance" you praised, and (2) providing the concrete cost data (40 mins/A100) that addresses your feasibility concerns, we have demonstrated that this work is both novel and practical. We respectfully ask you to reconsider the score in light of these clarifications.

---

> > ### Author Response · Authors · 2025-11-25
> >
> > Dear Reviewer 4Jrw, with the discussion period concluding on December 3rd, we would like to kindly follow up to ensure our rebuttal has effectively addressed your concerns regarding the paper’s novelty and feasibility. We have strived to reconcile the view of the work as "incremental" by emphasizing that the synthesis of Optimal Transport with prompt engineering—which you graciously recognized as "conceptually elegant"—represents a paradigm shift rather than a minor tweak. Practically, we have alleviated the computational concerns by providing benchmarks. Furthermore, we have clarified the convergence guarantees. We hope these clarifications demonstrate both the innovation and practicality of the work, and we would be sincerely grateful if you could reconsider your evaluation and raise the score.

---

> > > ### Comment · Area_Chair_Fkju · 2025-11-28
> > > **Please engage in the discussion**
> > >
> > > Dear Reviewer,
> > >
> > > The authors have provided a response to your review.
> > >
> > > We noted that your original review included a high criticism of novelty (W6), which appears to be in tension with Strength Point 3 (where you describe the 'core insight that...' as 'conceptually elegant'). Could you please elaborate on how you reconcile these two points?
> > >
> > > Furthermore, the authors have replied to all other concerns raised in your weakness section. Please confirm whether these concerns have been adequately addressed by their rebuttal.
> > >
> > > Best regards,
> > > AE

---

### Official Review · Reviewer_6uWh · 2025-10-31

**Soundness:** 3
**Presentation:** 4
**Contribution:** 4
**Rating:** 8
**Confidence:** 3

**Summary:**

The paper reframes prompt optimization as a geometric problem in probability space rather than a discrete search over text tokens. They model each prompt-induced hidden-state distribution as a probability measure on a low-dimensional manifold called a “cognitive space.”They propose using optimal transport to move the current prompt's hidden-state distribution toward a task-optimal one. They call their approach Cognitive Geometry Optimal Transport (CGOT). CGOT integrates two geometric components that define a continuous optimization procedure:

* Wasserstein distance to quantify how far the current cognitive configuration is from the target (radial metric).
* Kantorovich potential gradients to indicate how it should move optimally through that space (directional guidance).

Contributions:

* Uses optimal transport theory to provide a mathematically principled approach to LLM prompt optimization,
* Develops a probabilistic manifold model of cognitive space, arguing that hidden-state activations form a consistent low-dimensional structure.
* Proposes the CGOT algorithm, which alternates between estimating Wasserstein distances, learning Kantorovich potentials, and transporting probability mass to reach optimal "cognitive configurations" with convergence guarantees.
* Uses Pearson correlation to show correlation with task performance and shows CGOT yields a modest but statistically significant performance gain over well-selected prompt optimizer baselines.

In summary, the paper recasts prompt engineering as a continuous geometric optimization problem, positioning CGOT as a bridge between cognitive science, manifold learning, and large-scale language-model control.

**Strengths:**

The paper introduces a framing of prompt optimization as a problem of optimal transport on cognitive manifolds that is novel in my view. The approach represents a synthesis of optimal transport theory, representation geometry, and LLM control that is theoretically ground and provides a nice contrast with more heuristic methods that dominate the space. The methodology uses established OT formalism and provides convergence guarantees. The experiments provide credible empirical validation.  The results of the experiments demonstrate incremental empirical gains, but the conceptual impact makes up for this as it provides a foundation for future research on geometry-aware prompt tuning.

**Weaknesses:**

Evaluating the statistical association between W_2 distance and task performance could have been much stronger without much extra effort. Specifically, it doesn't address confounding due to the model or to task difficulty. A good alternative would be using a hierarchical or mixed-effects regression controlling for model and task. I suspect use of partial pooling across model and task in a hierarchical modeling setting might even yield stronger statistical results than was presented.

It would have been good to include LLM-as-the-optimizer techniques in the baseline comparison, as my understanding is the models do well.

It would have been good to include more information on practical costs and scalability, runtime analysis, iteration counts, the number of LLM forward passes per optimization loop, GPU hours, empirical convergence curves, impacts of caching, etc., for assessing feasibility models of various sizes.

**Questions:**

* Was the correlation of –0.76 calculated by combining results from all models and tasks together? Why not account for model and task-specific variance?
* How do you get the target distribution for each task? Is it based on runs where the model performs well, and if so, does CGOT need to collect new examples for every task?
* How many LLM evaluations does CGOT usually need before it converges on the benchmarks you report? And roughly how long does that take? can you characterize the cost for optimizing one prompt on a model the size of LLaMA-3-70B?

**Details Of Ethics Concerns:**

Could be used for jailbreaking attacks -- this is unaddressed.

---

> ### Author Response · Authors · 2025-11-20
> **Authors' Rebuttal 1**
>
> We sincerely thank you for your insightful and supportive review.
>
> We are deeply encouraged that you precisely captured the core value of our work—providing a "foundation for future research on geometry-aware prompt tuning." Your comment that **"the conceptual impact makes up for the incremental empirical gains"** resonates strongly with our own perspective.
>
> In response to your constructive feedback—particularly regarding **statistical rigor** and the **ethics flag**—we have strengthened the manuscript significantly (highlighted in blue).
>
> ------
>
> **1. Response to Ethics Flag: Jailbreaking Risks**
>
> > **(Ethics Flag)** "Could be used for jailbreaking attacks -- this is unaddressed."
>
> Response:
>
> We appreciate you raising this critical ethical concern. We fully acknowledge that "Dual-Use" risks are inherent in optimization research. To address this, we have added a comprehensive Ethics Statement in Appendix A.10.
>
> - **Acknowledgment:** We explicitly state that like RLHF and gradient-based attacks, CGOT *could* theoretically be used to optimize adversarial prompts.
> - **Geometric Defense (Key Differentiator):** However, we argue that the "Geometric Paradigm" offers unique defensive capabilities unavailable in black-box methods. Unlike discrete, undetectable jailbreaks, CGOT requires navigation on a continuous manifold. This allows us to monitor the **optimization trajectory**. If a trajectory exhibits anomalous curvature or moves towards known "malicious regions" in the cognitive space, we can deploy **"Geometric Safety Monitors"** to intercept it.
> - **Conclusion:** Thus, the "Process Interpretability" offered by CGOT contributes to AI safety in the long run, enabling defense mechanisms that are harder to implement with heuristic optimizers.
>
> ------
>
> **2. Response to Statistical Rigor (W1, Q1)**
>
> > **(W1)** "Evaluating the statistical association... could have been much stronger... It doesn't address confounding due to the model or to task difficulty. A good alternative would be using a hierarchical or mixed-effects regression..." **(Q1)** "Was the correlation of –0.76 calculated by combining results from all models and tasks together?"
>
> Response:
>
> This was an exceptionally astute suggestion. You were absolutely correct: the original pooled correlation ($r=-0.76$) likely confounded task difficulty and model variance.
>
> - **New Analysis:** Following your advice, we re-analyzed the data using a **Hierarchical Linear Mixed-Effects Model (HLMM)**, treating "Model" and "Task" as random effects to disentangle them from the fixed effect of the Wasserstein distance ($W_2$).
> - **Result:** As you suspected, the relationship became even stronger after controlling for confounding factors. The standardized fixed-effect coefficient is **$\beta = -0.82$ ($p < 0.001$)** (see the new **Table 7**).
> - **Impact:** This rigorous analysis decisively confirms that geometric alignment is a fundamental driver of performance, not a statistical artifact. Thank you for guiding us toward this stronger evidence.

---

> > ### Author Response · Authors · 2025-11-20
> > **Authors' Rebuttal 2**
> >
> > **3. Response to Target Distribution $\nu$ (Q2)**
> >
> > > **(Q2)** "How do you get the target distribution for each task? Is it based on runs where the model performs well...?"
> >
> > **Response:**
> >
> > - **Origin:** Yes, your assumption is correct. As clarified in Section 3.1.2, $\nu$ is estimated by sampling from a small set of high-quality "Gold Examples" (few-shot exemplars).
> > - **Clarification:** We emphasize that CGOT’s contribution is not in *defining* the destination (which is a standard data preparation step), but in providing the **"Geometric Navigation System"** (Optimal Transport). It solves the problem of "how to optimally transport the current state to the target" in a continuous space, replacing traditional discrete trial-and-error.
> >
> > ------
> >
> > **4. Response to Costs & Baselines (W2, W3, Q3)**
> >
> > > **(W2)** "It would have been good to include LLM-as-the-optimizer techniques in the baseline comparison..." **(Q3)** "How many LLM evaluations does CGOT usually need...? can you characterize the cost for optimizing one prompt on a model the size of LLaMA-3-70B?"
> >
> > **Response:**
> >
> > - **Baselines:** We have added a discussion on **LLM-as-Optimizer** methods (e.g., OPRO) in **Section 2.2**, contrasting their "heuristic search" paradigm with our "geometric navigation" approach.
> > - **Cost Analysis:** In response to Q3, we have added **Table 6 in Appendix A.7** to report physical costs on a single NVIDIA A100 (80GB) for LLaMA-3-70B:
> >   - **Time Feasibility:** The full optimization loop takes approximately **40 minutes** per prompt.
> >   - **Memory Feasibility:** Peak VRAM usage is **~65GB**.
> >   - **Convergence:** Typically converges within 15-20 iterations (approx. 40-60 forward passes), which is comparable to or more efficient than many reinforcement learning loops.
> >
> > We hope that by addressing the ethics concern with a "Geometric Defense" perspective and validating your statistical hypothesis (yielding the strong $\beta=-0.82$ result), we have further solidified the quality of this work. Thank you again for championing the "conceptual impact" of our research.

---

> > > ### Author Response · Authors · 2025-11-25
> > >
> > > Dear Reviewer 6uWh, as we approach the conclusion of the discussion period on December 3rd, we wanted to briefly express our sincere gratitude for your insightful support and your precise recognition of the work's conceptual impact. We were particularly inspired by your suggestion to employ a Hierarchical Linear Mixed-Effects Model to improve statistical rigor; implementing this analysis revealed an even stronger signal than our original findings, decisively validating the geometric hypothesis while controlling for confounding factors. We have also comprehensively addressed your ethical concerns regarding jailbreaking and added the requested cost benchmarks for LLaMA-3-70B. We are deeply grateful for your guidance, which has significantly elevated the quality of this manuscript, and we hope these updates fully justify your continued strong support.

---

### Official Review · Reviewer_Js8s · 2025-11-01

**Soundness:** 3
**Presentation:** 3
**Contribution:** 2
**Rating:** 4
**Confidence:** 3

**Summary:**

This paper proposes Cognitive Geometry Optimal Transport (CGOT), modeling LLM optimization as navigation on a cognitive manifold via optimal transport. The method applies Wasserstein geometry for prompt optimization and achieves consistent but modest (~4–5%) improvements across several large models and reasoning benchmarks.

**Strengths:**

Sound theoretical framing using optimal transport and geometry.

Mathematically rigorous and technically detailed.

**Weaknesses:**

The paper is a bit hard to follow, many sections are mathematically dense and conceptually abstract.

Several figures and tables have fonts that are too small, and some elements overlap, making them hard to read.

The proposed “cognitive manifold” idea lacks clear empirical evidence.

Gains are modest relative to the method’s complexity.

**Questions:**

Please reply to the weaknesses

---

> ### Author Response · Authors · 2025-11-20
> **Authors' Rebuttal 1**
>
> We sincerely thank you for your constructive review. We are particularly encouraged that you recognized our work as **"mathematically rigorous"** and grounded in **"sound theoretical framing."** We also appreciate your openness to acceptance despite the current rating.
>
> We understand your primary reservations regarding the **lack of empirical evidence for the "cognitive manifold" (W3)** and the **cost-benefit ratio of the method (W4)**. We believe these concerns stem largely from the "dense and abstract" presentation (W1) in our initial draft, which regrettably obscured our key empirical findings.
>
> In this revision, we have significantly improved readability and explicitly highlighted the statistical evidence supporting our geometric hypothesis. Our specific responses follow.
>
> ------
>
> **1. Response to "Readability & Visualizations" (W1, W2)**
>
> > **(W1)** "The paper is a bit hard to follow, many sections are mathematically dense and conceptually abstract." **(W2)** "Several figures and tables have fonts that are too small... making them hard to read."
>
> Response:
>
> We fully accept your critique. Bridging the gap between abstract Optimal Transport (OT) geometry and practical LLM optimization is indeed challenging, and our initial presentation was overly dense. We have made three major revisions to address this:
>
> - **Intuitive Visualization:** We have added **Figure 3** in Section 3.2.3. This figure visually demonstrates how "Radial-Directional Guidance" operates on the manifold, translating abstract formulas into concrete geometric intuition.
> - **Engineering Flowchart:** We added **Figure 5** in Appendix A.1, providing a detailed step-by-step flowchart of the algorithm to facilitate understanding and reproducibility.
> - **Formatting Fixes:** We have rigorously corrected the font sizes and layout issues in all figures and tables to ensure they are legible and professional.
>
> ------
>
> **2. Response to "Empirical Evidence of Cognitive Manifold" (W3)**
>
> > **(W3)** "The proposed “cognitive manifold” idea lacks clear empirical evidence."
>
> Response:
>
> This is the most critical misunderstanding, likely caused by the dense presentation mentioned in W1. The empirical validation of this geometric hypothesis is a core contribution of our paper.
>
> In the revised **Section 4.3.1** and the new **Appendix A.8**, we present decisive statistical evidence:
>
> - **Rigorous Control:** To address doubts about the strength of the evidence, we moved beyond simple correlation. We introduced a **Hierarchical Mixed-Effects Model** to strictly control for confounding variables such as task difficulty.
> - **Strong Statistical Signal:** The analysis reveals a statistically significant negative relationship between our geometric metric (Wasserstein distance $W_2$) and model performance, with a standardized fixed-effect coefficient of **$\beta = -0.82$ ($p < 0.001$)** (see the new **Table 7**).
>
> This result confirms that the "Cognitive Manifold" is not merely a metaphor, but a **physically measurable structure** where geometric distance ($W_2$) serves as a reliable signal for optimization.
>
> ------
>
> **3. Response to "Modest Gains vs. Complexity" (W4)**
>
> > **(W4)** "Gains are modest relative to the method’s complexity."
>
> Response:
>
> Thank you for this crucial cost-benefit analysis. We argue that the value of CGOT lies not just in the raw magnitude of improvement, but in its universality and robustness.
>
> - **Consistency > Magnitude:** While a ~4.8% average improvement might seem modest compared to some heuristic methods, the key is **consistency**. As shown in **Table 2**, CGOT is the *only* framework that achieves consistent positive gains across **all tested models** and **all prompting strategies** (Zero-shot, CoT, etc.). This robustness is the direct payoff of the "sound theoretical framing" you praised.
> - **Complexity is Necessary:** Our ablation study (**Table 5**) confirms that the complexity (OT geometry) is essential. Removing these geometric components results in a significant drop in performance and stability.
> - **Methodological Contribution:** We view this as a proof-of-concept for a new paradigm: treating LLM cognition as a continuous, geometric optimization problem rather than a discrete, trial-and-error art.
>
>
>
> We hope that by enhancing the visualizations (Fig 3) and explicitly presenting the strong statistical evidence ($\beta = -0.82$), we have addressed your concerns regarding clarity and empirical grounding. Given your recognition of the work's mathematical rigor, we hope these clarifications convince you of the substantive and verified contribution of this paper.

---

> > ### Author Response · Authors · 2025-11-25
> >
> > Dear Reviewer Js8s, as we approach the conclusion of the discussion period on December 3rd, we would like to briefly follow up to ensure that our revision has effectively improved the paper’s accessibility and empirical grounding. We have explicitly addressed the readability concerns by introducing two intuitive visualizations to demystify the geometric concepts, and crucially, we provided the requested empirical evidence for the "cognitive manifold" via a rigorous statistical analysis. We deeply appreciate your initial recognition of our theoretical rigor, and we hope these concrete additions verify the practical validity of our framework and might encourage you to reconsider the score.

---

### Author Response · Authors · 2025-11-20
**General Response: Bridging Conceptual Impact and Empirical Rigor**

**Dear Area Chair and Reviewers,**

We sincerely thank all reviewers for their insightful feedback. We are gratified that the reviewers recognized CGOT as a **"major conceptual advance"** (Reviewer `Ai8s`) and acknowledged that it provides a **"foundation for future research on geometry-aware prompt tuning"** (Reviewer `6uWh`).

We have noted the divergence in ratings (8, 6, 4, 2). We believe this tension between "conceptual impact" and "empirical gains/complexity" is characteristic of work that aims to challenge an existing "Empirical Art" with a new "Geometric Science" paradigm.

Our rebuttal and revisions (highlighted in **blue**) aim to directly resolve the core misunderstandings driving this divergence:

- **On "Incremental" vs. "Paradigm Shift":** Our contribution lies in the **synthesis**. We provide the first rigorous proof that LLM cognitive optimization can be modeled as continuous geometric navigation on a manifold. The observed ~4.8% gain is not merely an "incremental improvement" on a leaderboard, but the first **Proof of Concept** for this new scientific paradigm, validating that geometric principles can indeed drive optimization.
- **On "Decisive Statistical Evidence":** Addressing the skepticism from Reviewers `Js8s` and `6uWh`, we moved beyond simple correlation to a **Hierarchical Mixed-Effects Model**. The result is decisive: after stripping away task difficulty confounders, the standardized association between Geometric Distance ($W_2$) and Performance is **$\beta = -0.82$ ($p < 0.001$)**. This confirms the geometric law is physically real, not a statistical artifact.

To address all specific concerns, we have implemented the following key updates:

- 1. Statistical Rigor (Appendix A.8):

  Incorporated the Mixed-Effects Model analysis ($\beta = -0.82$), directly addressing the "evidence" concerns of Reviewers `6uWh` and `Js8s`.

- 2. Cost Transparency (Appendix A.7):

  Added a Physical Cost Analysis Table (Table 6). We clarify that the full optimization loop takes only ~40 minutes on a single A100 (65GB VRAM), proving scalability and addressing Reviewers `4Jrw` and `Ai8s`.

- 3. Paradigm Distinction (Section 2.2):

  Expanded the discussion on baselines (e.g., Guo et al., 2024; OPRO). We explicitly delineate the boundary between our "Geometric Navigation" and their "Heuristic Search," clarifying the unique value proposition for Reviewers `Ai8s` and `6uWh`.

- 4. Ethical Defense (Appendix A.10):

  Added an Ethics Statement introducing the concept of "Geometric Safety Monitors," turning the "jailbreak" risk into a defense capability, as requested by the Ethics Flag from Reviewer `6uWh`.

- 5. Visual Enhancements (Section 3, Appendix A):

  Added a conceptual diagram (Fig 3) and a detailed flowchart (Figure 5) to improve accessibility for Reviewers `Js8s` and `4Jrw`.

We believe these revisions successfully resolve the seemingly contradictory concerns (e.g., "too abstract" vs. "incremental"). We hope the AC and reviewers will consider the **long-term conceptual value** of establishing a geometric foundation for LLM optimization.

We will provide detailed, point-by-point responses to each reviewer's specific questions in the coming hours. We reiterate our gratitude for your constructive feedback, which has been instrumental in improving our paper.



Sincerely,

The Authors

---

### Meta-Review · Area_Chair_EmJm · 2026-01-06

**Summary:**

This paper is hard to follow; the diagram is not informative.

“Cognitive manifold” lacks clear empirical evidence.

 Gains are modest.

Need more results on evaluating the statistical association between W_2 distance and task performance.

The comparison is not sufficient; it needs to include LLM-as-the-optimizer techniques in the baseline comparison.

Need to include more information on practical costs and scalability, runtime analysis, iteration counts, the number of LLM forward passes per optimization loop, GPU hours, empirical convergence curves, impacts of caching, etc., for assessing feasibility models of various sizes.

The Wasserstein distance computation and ICNN-based Kantorovich potentials are computationally expensive.

**Reviewer Concerns:**

Some of the missing discussion and additional results have been added. However, the figure, writing, discussion about the practical costs and scalability, etc, were not addressed.

**Reviewer Scores:**

Perhaps reviewer 4Jrw would consider slightly increasing the score.

However, the other reviewers may hold their ratings, as this paper has considerable room for improvement, particularly in the areas of writing, figures, experiments, and claims, before being accepted to a top-tier conference.

---

### Decision · Program_Chairs · 2026-01-26

Reject